# TAS: Distilling Arbitrary Teacher and Student via a Hybrid Assistant

## Abstract

Most knowledge distillation (KD) methodologies predominantly focus on teacher-student pairs with **similar architectures**, such as both being convolutional neural networks (CNNs). However, the potential and flexibility of KD can be greatly improved by expanding it to novel Cross-Architecture KD (CAKD), where the knowledge of homogeneous and **heterogeneous** teachers can be transferred flexibly to a given student. The primary challenge in CAKD lies in the substantial feature gaps between heterogeneous models, originating from the distinction of their inherent inductive biases and module functions. To this end, we introduce an assistant model as a bridge to facilitate smooth feature knowledge transfer between heterogeneous teachers and students. More importantly, within our proposed design principle, the assistant model combines the advantages of cross-architecture inductive biases and module functions by merging convolution and attention modules derived from both student and teacher module functions. Furthermore, we observe that heterogeneous features exhibit diverse spatial distributions in CAKD, hindering the effectiveness of conventional pixel-wise mean squared error (MSE) loss. Therefore, we leverage a spatial-agnostic InfoNCE loss to align features after spatial smoothing, thereby improving the feature alignments in CAKD. Our proposed method is evaluated across some homogeneous model pairs and arbitrary heterogeneous combinations of CNNs, ViTs, and MLPs, achieving state-of-the-art performance for distilled models with a maximum gain of 11.47% on CIFAR-100 and 3.67% on ImageNet-1K. Our code and models will be released.

## 1 Introduction

Knowledge Distillation (KD) (Hinton et al., 2015; Romero et al., 2015) has been demonstrated as a powerful method to transfer knowledge from a pre-trained and cumbersome teacher model to a compact and efficient student model. Compared to the model trained from scratch, the performance of the student model distilled by appropriate teachers usually improves significantly. Commonly, knowledge transferred is derived from either the output logits (logits-based KD (Sun et al., 2024)) or the intermediate features (feature-based KD (Romero et al., 2015)) of the teacher model. Therefore, it is intuitive to understand different teachers have different knowledge (logits or features) determined by their unique architectures (Liu et al., 2021a).

Most existing KD approaches focus on similar-architecture distillation (Romero et al., 2015; Tian et al., 2020; Liu et al., 2023) (called SAKD), *i.e.*, optional teachers are restricted to a limited scope with structures similar to the student model. However, this homogeneous distillation presents two principal limitations: **(1) Limited Potential:** Compared to the broader range of arbitrary teachers (including homogeneous and heterogeneous ones), the restricted scope of teachers in SAKD may fail to include the optimal knowledge necessary to enhance the performance of certain students. For instance, as OFA (Hao et al., 2023) demonstrated, distilling knowledge from a heterogeneous ViT-Base to ResNet50 yields superior student performance compared to using a ResNet152 as the homogeneous teacher. **(2) Limited Flexibility:** The emergence of new models (Liu et al., 2022; Tolstikhin et al., 2021) or the scarcity of perfectly tuned homogeneous teachers in domain-specific tasks (Ronneberger et al., 2015; Li et al., 2024) poses significant challenges in obtaining suitable homogeneous teachers, thereby impeding the applicability of SAKD. Thus, this paper expands KD to cross-architecture KD (CAKD), investigating methods to distill knowledge from both homogeneous and heterogeneous teachers to students (Hao et al., 2023). By broadening the pool of optional

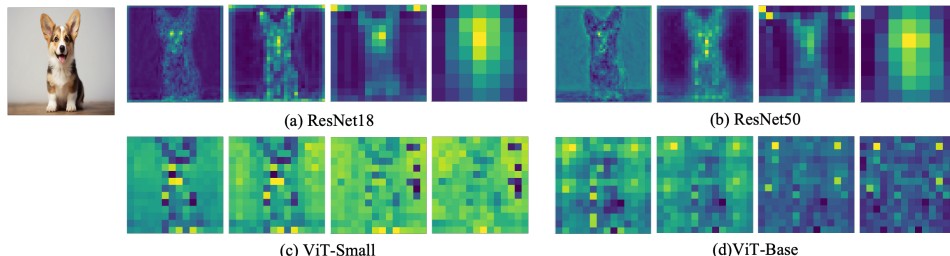

(a) ResNet18        (b) ResNet50

(c) ViT-Small        (d)ViT-Base

Figure 1: **Chanllenges.** In heterogeneous distillation, different models have different features in different stages caused by different inductive biases and module functions.

teachers, CAKD improves the potential and flexibility of single-teacher (Hinton et al., 2015) and multi-teacher (Liu et al., 2021a; Cao et al., 2023) KD compared to existing SAKD.

In CAKD, the main challenge is that heterogeneous teachers and students have significant representation gaps as detailed in feature analyses based on CKA alignment in OFA (Hao et al., 2023). These gaps stem from inherent differences in (1) inductive biases (Raghu et al., 2021) and (2) module functions (Liu et al., 2023). **(1) Inductive biases:** As depicted in Fig. 3 (a), convolutional-neural networks-based models (CNNs) (He et al., 2016; Sandler et al., 2018) exhibit hard inductive biases, specifically locality and translation-equivariance. Consequently, CNN-generated features are derived from local pixels within local receptive fields in Fig. 1 (a,b). In contrast, as shown in Fig. 3(b,c), multi-head-self-attention-based models (MSAs) (Dosovitskiy et al., 2021) and multilayer-perception-based models (MLPs) (Tolstikhin et al., 2021) have soft inductive biases (Bachmann et al., 2024; Park & Kim, 2021; Raghu et al., 2021), *i.e.*, patchify and long-distance dependency. Hence, as shown in Fig. 1(c,d) and Appendix C, features of most MSA/MLP models are generated by exchanging globally the information of all patches (Raghu et al., 2021). **(2) Module functions:** Varied module functions generate different features at different stages. For instance, features of shallow and deep layers in ViT have higher similarity than hierarchical CNN (Park & Kim, 2021; Raghu et al., 2021) in Fig. 1(a,c). Therefore, heterogeneous T.-S. pairs in CAKD exhibit significantly different inductive biases and module functions compared to SAKD, leading to substantial representation gaps that impede effective feature transfer. While OFA (Hao et al., 2023) attempts to address heterogeneous feature gaps by projecting features into the logit spaces, it is suboptimal due to substantial damage to feature-specific knowledge. This dilemma leads to a natural question: *Can we get the best use of different inductive biases and module functions, thereby reducing heterogeneous representation gaps and making feature transfer promising in CAKD?*

To alleviate heterogeneous feature gaps in CAKD, we introduce a Teacher-Assistant-Student distillation paradigm (T.-A.-S.[1], called TAS) by incorporating a hybrid assistant model as a bridge to facilitate smoother knowledge transfer. More importantly, our assistant model adheres to a novel design principle: *"the assistant model should be composed of merging CNN and MSA/MLP modules derived from teacher and student models"*.

Our design is well-motivated by the following popular beliefs: **(1) Why use a hybrid model as our assistant, not a pure CNN/MSA/MLP model?** As demonstrated in (Park & Kim, 2021; Li et al., 2023a;b), CNNs and MSAs/MLPs are complementary. A hybrid model that uses CNNs in the early stages and MSAs/MLPs in the later stages can benefit from both local and global inductive biases. Compared to existing KD like Fig. 2 (a-e), our proposed hybrid assistant model in Fig. 2 (f) projects heterogeneous features to a common space by merging CNN and MSA/MLP modules, thereby reducing distillation gaps attributed to inductive biases. **(2) Why is our assistant model composed of student and teacher modules, not an externally introduced hybrid model (Li et al., 2023b)?** As demonstrated in (Liu et al., 2023; Chen et al., 2021; 2022b), the disparity between heterogeneous features is also from module functions, *i.e.*, how the models will read, decode, and process the input features. Unlike an externally introduced model, a hybrid assistant model comprising student and teacher modules not only optimizes the functional similarity (Liu et al., 2023) between heterogeneous T.-S. pairs, but also introduces minimal additional learnable parameters in Appendix G. **(3) How do we align heterogeneous features spatially?** Widely used mean square error loss (MSE) aligns the features pixel-by-pixel, which is suitable for features that have similar spatial distribution,

---

[1]In this paper, teacher, assistant, and student model are shortened by T., A. and S..

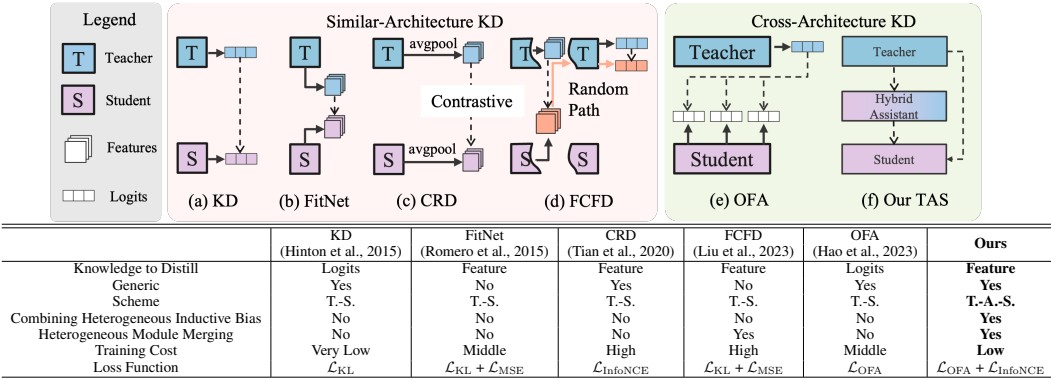

| | KD (Hinton et al., 2015) | FitNet (Romero et al., 2015) | CRD (Tian et al., 2020) | FCFD (Liu et al., 2023) | OFA (Hao et al., 2023) | **Ours** |
|---|---|---|---|---|---|---|
| Knowledge to Distill | Logits | Feature | Feature | Feature | Logits | **Feature** |
| Generic | Yes | No | Yes | No | Yes | **Yes** |
| Scheme | T.-S. | T.-S. | T.-S. | T.-S. | T.-S. | **T.-A.-S.** |
| Combining Heterogeneous Inductive Bias | No | No | No | No | No | **Yes** |
| Heterogeneous Module Merging | No | No | No | Yes | No | **Yes** |
| Training Cost | Very Low | Middle | High | High | Middle | **Low** |
| Loss Function | $\mathcal{L}_{\mathrm{KL}}$ | $\mathcal{L}_{\mathrm{KL}} + \mathcal{L}_{\mathrm{MSE}}$ | $\mathcal{L}_{\mathrm{InfoNCE}}$ | $\mathcal{L}_{\mathrm{KL}} + \mathcal{L}_{\mathrm{MSE}}$ | $\mathcal{L}_{\mathrm{OFA}}$ | $\mathcal{L}_{\mathrm{OFA}} + \mathcal{L}_{\mathrm{InfoNCE}}$ |

Figure 2: **The taxonomy of our method.** Our methods are feature-based, generic, and three-level, combining heterogeneous inductive biases and module functions with an efficient assistant. Target-wise $\mathcal{L}_{\mathrm{OFA}}$ and spatial-agnostic $\mathcal{L}_{\mathrm{InfoNCE}}$ are more suitable for CAKD than $\mathcal{L}_{\mathrm{KL}}$ and $\mathcal{L}_{\mathrm{MSE}}$. To the best of our knowledge, our TAS is the first feature-based generic distillation for arbitrary T.-S. pairs.

*e.g.*, features of ResNet18 *vs.* ResNet50 in Fig. 1(a-b). However, it is inadequate for spatially diverse features of heterogeneous models, *e.g.*, (a) and (c) in Fig. 1 show distinct completely spatial distributions at any stages. To address this, we first apply average pooling to smooth the spatial information of features and utilize a spatial-agnostic contrastive loss (InfoNCE (He et al., 2020; Chen et al., 2020; Tian et al., 2020)) to align heterogeneous feature embeddings.

In view of the above analysis, the taxonomy of our methods in KD is illustrated in Fig. 2. Our TAS falls under the category of *feature-based methods* for *generic distilltion* with a *hybrid-assistant* scheme. In our experiments, the proposed TAS greatly enhances the performance of student models in both CAKD and SAKD, achieving a maximum gain of 11.47% on the CIFAR100 and 3.67% on the ImageNet-1K, while maintaining a lower training cost compared to the SOTA (Hao et al., 2023).

## 2 RELATED WORK

### 2.1 TAXONOMY OF OUR METHODS

As shown in Fig. 2, the majority of existing KD methodologies concentrate on homogeneous distillation by using a single projector (*e.g.*, single linear layer) to align the output logits (Hinton et al., 2015; Huang et al., 2022; Sun et al., 2024), intermediate features (Romero et al., 2015; Chen et al., 2021; Liu et al., 2023), feature embeddings (Tian et al., 2020), and module functions (Liu et al., 2023) of T.-S. pairs, thanks to the highly-similar features between homogeneous T.-S. pairs. However, they fall short in addressing the complexities of heterogeneous distillation, where the distinct features between heterogeneous T.-S. pairs pose significant challenges. Although OFA Hao et al. (2023) achieves consistent improvements for arbitrary T.-S. pairs, it does so at the expense of sacrificing feature information to logits. In this paper, our method dives into the nature of heterogeneous feature gaps (*i.e.*, caused by inductive bias and module functions) and introduces a hybrid assistant to facilitate smoother feature transfer between heterogeneous T.-S. pairs.

Additionally, several other works are pertinent to our method: (1) We note that some methods attempt to distill the knowledge between CNNs and MSAs (Zhao et al., 2023), but they are tailored to specific T.-S. pairs rendering them impractical for our arbitrary T.-S. CAKD. (2) Certain methods apply progressive distillation to transfer the knowledge via a middle model (Mirzadeh et al., 2020; Cao et al., 2023; Liu et al., 2021a), but they are progressive training strategies that are not training algorithms designed for transferring knowledge between heterogeneous T.-S. pairs. (3) Lastly, while some logits-based methods can be easily applied to CAKD (Hinton et al., 2015; Sun et al., 2024; Hao et al., 2023), they are suboptimal because they overlook the significance of feature-based knowledge (Romero et al., 2015). Conversely, our method (1) is easily applied arbitrary T.-S. pairs with two important design principles, (2) employs three-level joint optimization algorithms, and (3) utilizes a generic feature-based method for both homogeneous and heterogeneous distillation.

In this paper, our method focuses on feature-based CAKD with a three-level distilling paradigm.

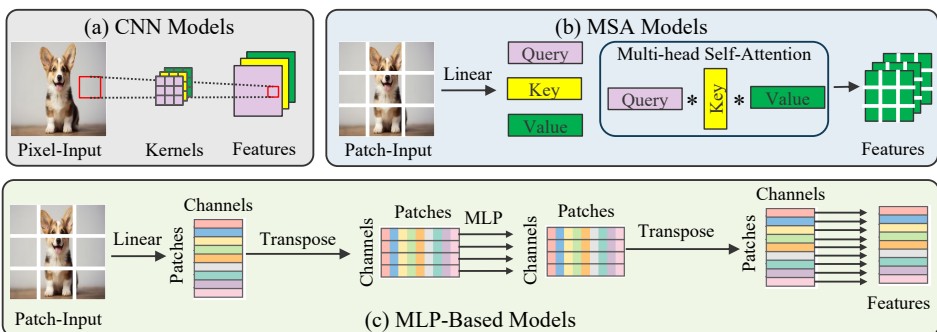

Figure 3: **Different inductive biases and module functions.** (a) CNN models (He et al., 2016) generate features with locality and translation-equivariance. (b) The features are generated by pacifying and global attention schemes in MSA models (Dosovitskiy et al., 2021). (c) Token- and channel-mixing MLP exchange feature information of global patches in MLP models.

## 2.2 HYBRID MODEL

As illustrated in Fig. 3, different models exhibit different inductive biases and module functions. (Raghu et al., 2021) investigates the internal representation structures of ViT and CNN models, revealing significant differences between their heterogeneous features. (Park & Kim, 2021) further provides some fundamental explanations for this phenomenon. Specifically, CNNs are data-agnostic and channel-specific high-pass filters, while MSAs are data-specific and channel-agnostic low-pass filters. Therefore, researchers (Park & Kim, 2021) think CNNs and MSAs are complementary, which inspires them to design a hybrid model following the rules of "alternately replacing CNN blocks with MSA blocks from the end of a baseline CNN model". The hybrid model outperforms CNNs in both large and small data regimes (Park & Kim, 2021). Furthermore, the architecture of MLP models (Tolstikhin et al., 2021) is notably similar to ViTs not CNNs, so ConvMLP (Li et al., 2023a) also achieves advanced performance in basic visual tasks by the co-design of CNNs and MLPs. In a nutshell, hybrid CNN-MSA/MLP models improve performance and efficiency through the combination of different inductive biases and module functions.

Inspired by the design of the hybrid model, we mitigate the heterogeneous feature gaps by introducing a hybrid assistant model between cross-architecture T.-S. pairs.

## 3 METHOD

### 3.1 PRELIMINARIES

Existing KD methods perform well in homogeneous distillation, but they may fail in heterogeneous teachers and students. The primary reasons stem from fundamentally distinct feature and logit spaces, caused by different *inductive biases* and *module functions* of heterogeneous models.

**Inductive Bias and module functions.** Inductive bias refers to the set of assumptions that a model uses to make predictions on unseen data (Ren et al., 2022). Module functions describe how a model reads, encodes, decodes, and processes the data (Liu et al., 2023). As shown in Fig. 3, heterogeneous models exhibit different inductive biases and module functions. (1) CNN models (He et al., 2016) slide a set of learnable local kernels across the pixel-level image, focusing on local receptive fields. The weight-sharing kernels are applied across the entire image, providing the network with translation-equivariance to recognize an object regardless of location. (2) MSA models (Dosovitskiy et al., 2021) split the input image into patches, which are then projected into Query, Key, and Value. The attention modules calculate the scores between the Query and Key to generate attention maps, which are then used to weight the corresponding Value. This process, capturing long-distance dependency, allows the model to consider the global information from all patches. (3) MLP models (Touvron et al., 2022) also begin by dividing the input image into patches. It then mixes global information along all patches' spatial and channel dimensions. In a nutshell, different inductive biases and module functions determine different distribution of generated features.

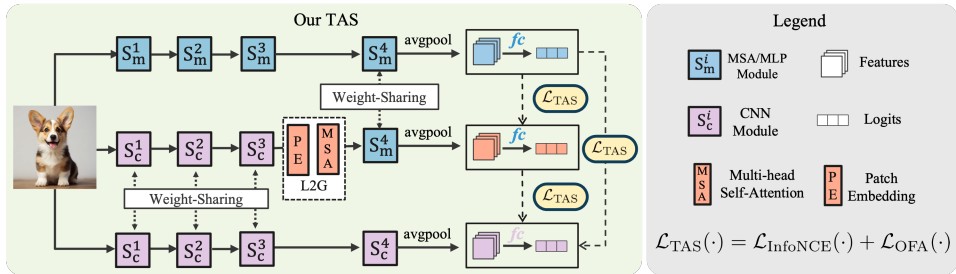

Figure 4: **Overall.** Our TAS applies CAKD by introducing a hybrid assistant model, which generates features by merging the first three stages of CNNs, a projector L2G, and the last stage of MSAs/MLPs, thereby aligning cross-architecture inductive biases and module functions. Due to the spatial gaps among heterogeneous features, we transfer knowledge of spatial-smoothed features and logits with T.-A.-S. scheme. All models are split into four stages following OFA (Hao et al., 2023).

## 3.2 THREE-LEVEL DISTILLATION PARADIGM

As shown in (a-e) of Fig. 2, existing methods usually apply a two-level paradigm in SAKD (Hinton et al., 2015; Romero et al., 2015; Liu et al., 2023), *i.e.*, T.-S. scheme, to transfer directly the knowledge of teachers to students. Besides, some works apply progressive training strategy (Mirzadeh et al., 2020; Liu et al., 2021a) to transfer multi-teacher knowledge to a single student model in SAKD, but they are also a two-level paradigm for each T.-S. distillation. Despite these two-level paradigms being better suited for SAKD, transferring knowledge directly without any transition mechanism is challenging for our CAKD due to the spatial diversity of features in Fig. 1.

In the human learning process, the role of teaching assistants greatly reduces the information gaps between teachers and students. Motivated by this, as illustrated in Fig. 4, this paper introduces an assistant model as a middle bridge, where the knowledge is transferred by training a teacher-assistant-student scheme (TAS) as follows:

$$\mathcal{L} = \mathcal{L}_{\text{TAS}}(\text{K}_\text{t}, \text{K}_\text{s}) + \mathcal{L}_{\text{TAS}}(\text{K}_\text{t}, \text{K}_\text{a}) + \mathcal{L}_{\text{TAS}}(\text{K}_\text{a}, \text{K}_\text{s}), \tag{1}$$

where $\mathcal{L}_{\text{TAS}}$ is our loss (details in Eq. (3)). $\text{K}_\text{t}$, $\text{K}_\text{a}$, and $\text{K}_\text{s}$ denote the knowledge of the teacher, assistant, and student model respectively.

**Assistant Model.** Our assistant model connects the CNN modules and MSA/MLP modules derived from the students and teachers with a local-to-global (L2G) feature projector as shown in Fig. 4. Formulary, our assistant model can be described as follows:

$$p_\text{a}(x) = fc_\text{m} \circ \text{S}_\text{m}^4 \circ (\text{MSA} \circ \text{PE}) \circ \text{S}_\text{c}^3 \circ \text{S}_\text{c}^2 \circ \text{S}_\text{c}^1(x), \tag{2}$$

where $x$ is the input image, $\text{S}_\text{c}^i$ denotes CNN models, $\text{S}_\text{m}^i$ denotes MSAs/MLPs, and $fc_\text{m}$ denotes the fully-connected layers of MSAs/MLPs. To connect CNN and MSA/MLP modules, we propose an L2G module that includes a patch embedding (Dosovitskiy et al., 2021) to convert the features into the required dimensions of subsequent MSA/MLP modules. Besides, to capture the long-distance dependency, our L2G also includes an MSA block to project the local features from CNN models to global receptive fields. For simplicity, the MSA module is a Swin block (Liu et al., 2021b) in this paper (more discussions in supplementary materials). Note that the L2G is the only extra learnable module in our assistant model.

The following considerations drive the design of our assistant model: (1) Inductive biases of CNNs and MSAs/MLPs are complementary and hybrid CNN-MSA/MLP models demonstrate good performance (Raghu et al., 2021; Park & Kim, 2021; Li et al., 2023a). Therefore, we replace the CNN blocks at the end of a baseline CNN model with MSA/MLP blocks as our assistant model which can benefit from both the local features in the early stages and the global information exchanges in the last stages. For example, final feature appearances of the assistant model are converted from the local CNN features to global receptive fields in Fig. 5. (2) Different models have different module functions, which can be aligned implicitly by connecting different module functions in a single pipeline (Liu et al., 2023). Therefore, we form our assistant model by using CNN/MSA/MLP modules derived from students and teachers. As shown in Fig. 4, our assistant model is mainly composed

of weight-sharing modules $S_c^{1 \to 3}$ and $S_m^4$, which not only unifies different module functions but also introduces negligible additional learnable parameters. Besides, weight-sharing modules also ensure the middle performance of our assistant model in Tab. 6. (3) While alternately using CNN and MSA modules can improve the performance of the hybrid assistant model (Park & Kim, 2021), we only add one stage of MSA/MLP models following the first three stages of CNN models as illustrated in Eq. (2) for simplicity (details discussed in Sec. 4.4).

**Loss Function.** As shown in Fig. 4, we only transfer the **final** features after **average pooling** and the logits for the following reasons. (1) Due to the weight-sharing between our assistant model and T.-S., it combines actually different inductive biases only in the final features, not early and middle features. (2) As shown in Fig. 5 and Fig. 1, the final features of different models are also very different in spatial, so we smooth them by average pooling to mitigate the spatial gaps between different features. The knowledge in Eq. (1) is formulated by $K_i = \{f_i, p_i\}$, $i = t, a, s$, where $f_i$ and $p_i$ denote the final features embeddings after average pooling and the output logits.

In this paper, we use spatial-agnostic InfoNCE loss $\mathcal{L}_{\text{InfoNCE}}$ (He et al., 2020; Tian et al., 2020) and OFA loss $\mathcal{L}_{\text{OFA}}$ to supervise the transfer of features and logits respectively, motivated by the following observations. (1) Wildly used MSE loss computes the pixel-wise metrics that are suitable for features having similar spatial information, but it will fail when the features are very spatially different (*e.g.*, FitNet with MSE loss gets only 24.06% top-1 accuracy when the teacher is ConvNeXt-T and the student is Swin-P in CIFAR100 Tab. 1). Consequently, we use a contrastive loss InfoNCE (He et al., 2020; Tian et al., 2020) to transfer the structural information of feature embeddings (Tian et al., 2020), which captures complex interdependencies of features without spatial information. (2) As demonstrated in (Hao et al., 2023), the different inductive bias leads models to variant logit spaces. For example, local CNN models are more suitable for small objects, but global MSA/MLP models are more suitable for large objects. Therefore, $\mathcal{L}_{\text{OFA}}$ enhances the information of the target class by adding a modulating parameter $\gamma$ to the original KD loss, which prevents the student learning from being disturbed by irrelevant information of the teacher. In a nutshell, our $\mathcal{L}_{\text{TAS}}$ is suitable for any representation distillation (*e.g.*, the superior consistently performance in Tab. 2, Tab. 1, and Tab. 3):

$$\mathcal{L}_{\text{TAS}}(K_t, K_s) = \begin{cases} \mathcal{L}_{\text{OFA}}(p_t, p_s) = (1 + p_t^{\hat{c}})^\gamma \log(\frac{p_t^{\hat{c}}}{p_s^{\hat{c}}}) + \sum\limits_{i=1, i \neq \hat{c}}^{C} p_t^c \log(\frac{p_t^c}{p_s^c}), \\ \mathcal{L}_{\text{InfoNCE}}(f_t, f_s) = -\log \frac{\exp(f_s \cdot f_t^+ / \tau_2)}{\sum_{i=0}^{F_t} \exp(f_s \cdot f_t^i / \tau_2)} \end{cases} \quad (3)$$

For each T.-S. pair, we transfer knowledge by $\mathcal{L}_{\text{TAS}}(K_t, K_s) = \mathcal{L}_{\text{InfoNCE}}(f_t, f_s) + \mathcal{L}_{\text{OFA}}(p_t, p_s)$. Firstly, for $\mathcal{L}_{\text{OFA}}$, the $\hat{c}$ and $c$ denote the target class and predicted class of the input image. $C$ is the all classes in the dataset. $\mathcal{L}_{\text{OFA}}$ add a modulating parameter $\gamma$ to enhance the target information when the teacher is not confident about the prediction. When $\gamma = 1$, $\mathcal{L}_{\text{OFA}}$ is equal to $\mathcal{L}_{\text{KD}}$ with temperate $\tau = 1$. Secondly, for $\mathcal{L}_{\text{InfoNCE}}$, $f_s$ denotes an encoded student features by average pooling, and $F_t$ is a set of encoded teacher features in a mini-batch. In $F_t$, only one positive sample $f_t^+$ matches to $f_s$, *i.e.*, the student's and teacher's feature from the same image is a positive pair. The InfoNCE loss is low when the features of student $f_s$ and teacher $f_t^+$ are from the same image and high when they are from different images. This loss has been widely demonstrated for aligning different feature representations (He et al., 2020; Tian et al., 2020).The temperature parameter $\tau_2$ is learnable (Radford et al., 2021). Lastly, the entire loss is $\mathcal{L} = \mathcal{L}_{\text{TAS}}(K_t, K_s) + \mathcal{L}_{\text{TAS}}(K_t, K_a) + \mathcal{L}_{\text{TAS}}(K_a, K_s)$, where the formulas of $\mathcal{L}_{\text{TAS}}(K_t, K_a)$ and $\mathcal{L}_{\text{TAS}}(K_a, K_s)$ is like $\mathcal{L}_{\text{TAS}}(K_t, K_s)$.

## 4 EXPERIMENTS

### 4.1 IMPLEMENTARY DETAILS

**Models.** For a fair comparison, we evaluate our TAS using the same teacher-student pairs employed in OFA(Hao et al., 2023), including homogeneous distillation and heterogeneous combinations of CNNs, MSAs, and MLPs. Specifically, CNN models include ResNet (He et al., 2016), MobileNetv2 (Sandler et al., 2018), and ConvNeXt (Liu et al., 2022). MSA models cover ViT, DeiT (Dosovitskiy et al., 2021; Touvron et al., 2021), and Swin (Liu et al., 2021b), while MLP

Table 1: **Top-1 accuracy (%) on CIFAR100.** All baseline results are from the paper or code of OFA (Hao et al., 2023). Swin-P is a modified version of Swin-T(Liu et al., 2021b) from OFA (Hao et al., 2023). **Bold** denotes the best results and the second-best results are underlined.

| Teacher | Student | From Scratch | | feature-based | | | | logits-based | | | CAKD | |
|---|---|---|---|---|---|---|---|---|---|---|---|---|
| | | Teacher | Student | FitNet | CC | RKD | CRD | KD | DKD | DIST | OFA | **TAS** |
| *CNN-based students* | | | | | | | | | | | | |
| Swin-T | ResNet18 | 89.26 | 74.01 | 78.87 | 74.19 | 74.11 | 77.63 | 78.74 | 80.26 | 77.75 | 80.54 | **81.61** |
| ViT-S | ResNet18 | 92.04 | 74.01 | 77.71 | 74.26 | 73.72 | 76.60 | 77.26 | 78.10 | 76.49 | 80.15 | **81.93** |
| Mixer-B/16 | ResNet18 | 87.29 | 74.01 | 77.15 | 74.26 | 73.75 | 76.42 | 77.79 | 78.67 | 76.36 | 79.39 | **81.90** |
| Swin-T | MobileNetV2 | 89.26 | 73.68 | 74.28 | 71.19 | 69.00 | 79.80 | 74.68 | 71.07 | 72.89 | 80.98 | **81.28** |
| ViT-S | MobileNetV2 | 92.04 | 73.68 | 73.54 | 70.67 | 68.46 | 78.14 | 72.77 | 69.80 | 72.54 | 78.45 | **82.10** |
| Mixer-B/16 | MobileNetV2 | 87.29 | 73.68 | 73.78 | 70.73 | 68.95 | 78.15 | 73.33 | 70.20 | 73.26 | 78.78 | **80.83** |
| *MSA-based students* | | | | | | | | | | | | |
| ConvNeXt-T | DeiT-T | 88.41 | 68.00 | 60.78 | 68.01 | 69.79 | 65.94 | 72.99 | 74.60 | 73.55 | 75.76 | **79.57** |
| Mixer-B/16 | DeiT-T | 87.29 | 68.00 | 71.05 | 68.13 | 69.89 | 65.35 | 71.36 | 73.44 | 71.67 | 73.90 | **74.40** |
| ConvNeXt-T | Swin-P | 88.41 | 72.63 | 24.06 | 72.63 | 71.73 | 67.09 | 76.44 | 76.8 | 76.41 | 78.32 | **80.73** |
| Mixer-B/16 | Swin-P | 87.29 | 72.63 | 75.2 | 73.32 | 70.82 | 67.03 | 75.93 | 76.39 | 75.85 | 76.65 | **78.44** |
| *MLP-based students* | | | | | | | | | | | | |
| ConvNeXt-t | ResMLP-S12 | 88.41 | 66.56 | 45.47 | 67.70 | 65.82 | 63.35 | 72.25 | 73.22 | 71.93 | 75.21 | **78.03** |
| Swin-T | ResMLP-S12 | 89.26 | 66.56 | 63.12 | 68.37 | 64.66 | 61.72 | 71.89 | 72.82 | 11.05 | 73.58 | **77.20** |
| Average Improvements | | | | −5.21 | −0.33 | −1.39 | −0.02 | +3.12 | +3.16 | −2.31 | +6.19 | **+8.38** |

models consist of MLP-Mixer (Tolstikhin et al., 2021) and ResMLP (Touvron et al., 2022). We averagely divide the models into 4 stages following OFA (Hao et al., 2023).

**Datasets.** We use the CIFAR100 (Krizhevsky et al., 2009) and ImageNet-1K dataset (Deng et al., 2009) for evaluation. CIFAR100 consists of 50K training samples and 10K testing samples in a resolution of 32×32, while the ImageNet-1K dataset contains 1.2 million training samples and 50K validation samples with a resolution of 224×224. Since MSAs and MLPs accept image patches as input, we upsample the images in CIFAR100 to the resolution of 224×224 (Hao et al., 2023).

**Baselines.** In line with OFA (Hao et al., 2023), we choose several powerful KD methods as our baselines for comparison. Specifically, the feature-based methods include FitNet (Romero et al., 2015), CC (Peng et al., 2019), RKD (Park et al., 2019), and CRD (Tian et al., 2020), while the logits-based methods comprise KD (Hinton et al., 2015), DKD (Zhao et al., 2022), and DIST (Huang et al., 2022). Originally, these methods were designed for SAKD, and thus OFA made some modifications to effectively apply them to CAKD scenarios.

**Training Protocols.** Following the OFA Hao et al. (2023), we utilize SGD optimizer for CNN-based students and AdamW optimizer for MSA- and MLP-based students. All models are trained for 300 epochs in the CIFAR100 dataset. As for the ImageNet-1K dataset, CNNs and MSA/MLP models are trained for 100 epochs and 300 epochs respectively. More details about training schedules and hyperparameters are in Appendix A.

## 4.2 MAIN RESULTS

Given extensive cross-architecture teacher-student model pairs, our TAS consistently achieves the best or most competitive performance on the CIFAR100 (+8.38% on average Top-1 accuracy) dataset and the ImageNet-1K dataset (+2.31% on average Top-1 accuracy).

**Results on CIFAR100.** To evaluate the performance in enough cross-architecture situations, as shown in Tab. 1, we conduct extensive experiments in 12 combinations of heterogeneous T.-S. models. We have the following important observations in this small-scale dataset.

Firstly, the feature-based methods exhibit inferior performance on most occasions, *e.g.*, they have negative performance on average improvements, especially when facing the MSA/MLP student models. The reason is that, as discussed in Sec. 3.1, the features of cross-architecture models are distinct as the different inductive bias and module functions, in which a naive feature projector struggles to address this dilemma in the case of small-scale datasets.

Secondly, FitNet (Romero et al., 2015) shows very poor performance when the teacher is ConvNeXt-T and the student is Swin-P, while the other feature-based and logits-based methods obtain relatively normal performance. We believe that this limitation of FitNet stems from its use of the MSE loss to align intermediate features of the student and teacher models in a pixel-wise manner, while other methods solely transfer knowledge from the final high-level feature embeddings or logits. In other

Table 2: **Top-1 accuracy (%) on ImageNet-1K.** All baseline results are from the paper or code of OFA (Hao et al., 2023). Swin-N is a modified version of Swin-T(Liu et al., 2021b) from OFA (Hao et al., 2023). **Bold** denotes the best results, and the second-best results are underlined.

| Teacher | Student | From Scratch | | feature-based | | | | logits-based | | | CAKD | |
|---|---|---|---|---|---|---|---|---|---|---|---|---|
| | | Teacher | Student | FitNet | CC | RKD | CRD | KD | DKD | DIST | OFA | **TAS** |
| *CNN-based models* | | | | | | | | | | | | |
| DeiT-T | ResNet18 | 72.17 | 69.75 | 70.44 | 69.77 | 69.47 | 69.25 | 70.22 | 69.39 | 70.64 | 71.01 | **71.22** |
| Swin-T | ResNet18 | 81.38 | 69.75 | 71.18 | 70.07 | 68.89 | 69.09 | 71.14 | 71.10 | 70.91 | 71.76 | **72.21** |
| Mixer-B/16 | ResNet18 | 76.62 | 69.75 | 70.78 | 70.05 | 69.46 | 68.4 | 70.89 | 69.89 | 70.66 | 71.38 | **71.44** |
| DeiT-T | MobileNetV2 | 72.17 | 68.87 | 70.95 | 70.69 | 69.72 | 69.6 | 70.87 | 70.14 | 71.08 | 71.39 | **71.78** |
| Swin-T | MobileNetV2 | 81.38 | 68.87 | 71.75 | 70.69 | 67.52 | 69.58 | 72.05 | 71.71 | 71.76 | 72.32 | **72.54** |
| Mixer-B/16 | MobileNetV2 | 76.62 | 68.87 | 71.59 | 70.79 | 69.86 | 68.89 | 71.92 | 70.93 | 71.74 | 72.12 | **72.31** |
| *MSA-based Models* | | | | | | | | | | | | |
| ResNet50 | DeiT-T | 80.38 | 72.17 | **75.84** | 72.56 | 72.06 | 68.53 | 75.10 | 75.6 | 75.13 | 75.73 | 75.64 |
| ConvNeXt-T | DeiT-T | 82.05 | 72.17 | 70.45 | 73.12 | 71.47 | 69.18 | 74.00 | 73.95 | 74.07 | 74.41 | **75.26** |
| Mixer-B/16 | DeiT-T | 76.62 | 72.17 | 74.38 | 72.82 | 72.24 | 68.23 | 74.16 | 72.82 | 74.22 | 74.46 | **75.00** |
| ResNet50 | Swin-N | 82.05 | 75.53 | 76.83 | 76.05 | 75.90 | 73.90 | 77.58 | 76.24 | 77.29 | 77.76 | **77.79** |
| ConvNeXt-T | Swin-N | 82.05 | 75.53 | 74.81 | 75.79 | 75.48 | 74.15 | 77.15 | 77.00 | 77.25 | 77.5 | **77.73** |
| Mixer-B/16 | Swin-N | 76.62 | 75.53 | 76.17 | 75.81 | 75.52 | 73.38 | 76.26 | 75.03 | 76.54 | 76.63 | **76.87** |
| *MLP-based models* | | | | | | | | | | | | |
| ConvNeXt-T | ResMLP-S12 | 82.05 | 76.65 | 74.69 | 75.79 | 75.28 | 73.57 | 76.87 | 77.23 | 77.24 | 77.26 | **77.33** |
| Swin-T | ResMLP-S12 | 81.38 | 76.65 | 76.48 | 76.15 | 75.1 | 73.4 | 76.67 | 76.99 | 77.25 | 77.31 | **77.42** |
| Average Improvements | | | | +1.00 | +0.56 | −0.30 | −1.65 | +1.61 | +1.05 | +1.65 | +2.05 | +2.31 |

Table 3: **Results in SAKD on ImageNet-1K.** The teacher and student are ResNet34 and ResNet18 in (a) and are ResNet50 and MobileNet in (b). As shown, our TAS method is still competitive in similar-architecture distillation.

| | T. | S. | AT | OFD | CRD | Review | DKD | DIST | FCFD | OFA | Ours |
|---|---|---|---|---|---|---|---|---|---|---|---|---|
| (a) | 73.31 | 70.66 | 70.69 | 70.81 | 71.17 | 71.61 | 71.70 | 72.07 | 72.24 | 72.10 | **72.29** |
| (b) | 76.61 | 68.58 | 69.56 | 71.25 | 71.37 | 72.56 | 72.05 | 73.24 | 73.37 | 73.28 | **73.45** |

words, applying pixel-wise MSE loss may not be suitable for spatially diverse feature maps of student and teacher models, as illustrated in Fig. 1 and Appendix E. Therefore, it is more suitable to transfer final features after smoothing spatial information, the same as our loss function in Eq. (3).

Lastly, OFA (Hao et al., 2023) yields significant and consistent improvements under all settings. However, these improvements come at the expense of structural feature information (Tian et al., 2020). For instance, as shown in Fig. 1(e), original feature dimensions exhibit complex interdependencies that would be damaged after projecting the features to logit spaces where each class is more independent (Tian et al., 2020). In contrast, our framework bridges the cross-architecture representation gaps via a hybrid assistant model and contrastive learning applied to spatial-smoothed features. Leveraging the two designs, our TAS achieves the best results in all T.-S. pairs in CAKD, obtaining an average gain of about 2.06% compared to the recent SOTA method OFA (Hao et al., 2023) on CIFAR100.

**Results on ImageNet-1K.** We also conduct extensive experiments on 14 combinations of cross-architecture T.-S. models on the large-scale ImageNet-1K dataset. Here, we observe that feature-based methods perform well when handling MSA/MLP students, for instance, utilizing Fit-Net (Romero et al., 2015) when the teacher model is ResNet50 and the student model is DeiT-T. This is opposite to our observations on CIFAR100. We argue that this discrepancy arises due to the data-hungry nature of MLP/MSA models and additional linear feature projectors (Park & Kim, 2021), which are better suited for training on large-scale datasets. Even so, traditional feature-based methods still have negative impacts in some other situations, *e.g.*, FitNet yields 70.45% (-1.72%) when the teacher is ConvNeXt-T and the student is DeiT-T. In a nutshell, even in training with large-scale data, simple feature projectors are not sufficient to transfer the features of students into arbitrary cross-architecture teacher spaces.

In this paper, besides the feature projectors L2G, our assistant model also includes modules derived from students and frozen teachers. In other words, our assistant model achieves a more important task, *i.e.*, aligning the knowledge of student functions to match the frozen teacher functions. Therefore, our TAS leverages more information and constraints than linear feature projectors in existing feature-based methods, leading to superior and stable performance on extensive combinations of cross-architecture models in the large-scale dataset. Besides, compared to leveraging four intermediate features of SOTA (Hao et al., 2023), our assistant achieves more competitive performance by only leveraging the final features.

Table 4: **Ablation study.** We evaluate the ablation studies by removing some important components of our assistant model and loss functions.

| Methods | CIFAR100 | | | | ImageNet | | | |
|---|---|---|---|---|---|---|---|---|
| | T. | S. | T. | S. | T. | S. | T. | S. |
| | Swin-T | ResNet18 | ConvNeXt-T | Swin-P | Swin-T | ResNet18 | ResNet50 | DeiT-T |
| KD (Baseline) | 78.74(-2.87) | | 76.44(-4.29) | | 71.14(-1.04) | | 75.10(-0.54) | |
| The architecture of assistant model | | | | | | | | |
| (A) w/o MSA and $S_m^4$ in Eq. (2) | 75.95(-5.66) | | 77.65(-3.18) | | 70.86(-1.35) | | 74.67(-0.97) | |
| (B) w/o $S_m^4$ in Eq. (2) | 77.21(-4.40) | | 77.84(-2.89) | | 71.78(-0.43) | | 75.14(-0.50) | |
| Loss functions | | | | | | | | |
| (C) w/o $\mathcal{L}_{\text{TAS}}(K_t, K_a)$ in Eq. (1) | 25.57(-56.04) | | 50.46(-30.27) | | 71.34(-0.87) | | 74.56(-1.08) | |
| (D) w/o $\mathcal{L}_{\text{TAS}}(K_a, K_s)$ in Eq. (1) | 79.01(-2.60) | | 79.82(-0.91) | | 71.46(-0.75) | | 74.81(-0.83) | |
| (E) w/o $\mathcal{L}_{\text{TAS}}(K_t, K_s)$ in Eq. (1) | 79.26(-0.66) | | 80.17(-0.56) | | 71.45(-0.76) | | 73.92(-1.72) | |
| (F) w/o $\mathcal{L}_{\text{InfoNCE}}$ in Eq. (3) | 79.28(-2.33) | | 78.89(-1.84) | | 71.47(-0.74) | | 75.21(-0.43) | |
| (G) w/o $\mathcal{L}_{\text{OFA}}$ in Eq. (3) | 77.91(-3.70) | | 80.32(-0.41) | | 70.37(-1.84) | | 72.13(-3.51) | |
| Ours | 81.61 | | 80.73 | | 72.21 | | 75.64 | |

**Results in SAKD.** As shown in Tab. 3, we compare the distilled results of AT (Chen et al., 2022a), OFD (Heo et al., 2019), CRD (Tian et al., 2020), Review (Chen et al., 2021), DKD (Zhao et al., 2022), DIST (Huang et al., 2022), FCFD (Liu et al., 2023) and OFA (Hao et al., 2023) on ImageNet-1k dataset. Compared to the recent works in homogeneous distillation (FCFD (Liu et al., 2023)) and heterogeneous distillation (OFA (Hao et al., 2023)), our TAS has a better performance when the teacher/student is ResNet34/ResNet18 and ResNet50/MobileNet. Therefore, our TAS is generic for any T.-S. pairs (including both homogeneous and heterogeneous pairs). Although the distilled performance of ResNet18 in SAKD is better than CAKD in Tab. 2, we find that CAKD still improves the distilled student after applying SAKD (multi-teacher progressive distillation (Mirzadeh et al., 2020), details in our Appendix D). In other words, given a student, heterogeneous teachers have different knowledge compared to homogeneous teachers, which can further improve students.

## 4.3 ABLATION STUDY

**The architecture of the assistant model.** In Tab. 4(A-B), we remove the module $S_m^4$ in (B) and the MSA module in (A), the performance of different teacher-student pairs drops significantly. This demonstrates the power of combining the inductive biases by adding MSA modules following the CNN modules and combining the module functions by adding $S_m^4$. Besides, different MSA blocks have different functions for different T.-S. pairs (details in Appendix F), so we use the Swin block as our MSA block in L2G for simplicity.

**The components of the three-level paradigm.** (1) In Tab. 4 (C), we remove the supervision from the teacher to our assistant model, *i.e.*, $\mathcal{L}_{\text{TAS}}(K_t, K_a)$, which makes the assistant model learn no correct knowledge from the teachers and then transfers the incorrect knowledge to the students. So the distilled students have poor performance. (2) We remove the supervision from the assistant model to students in Tab. 4 (D), *i.e.*, $\mathcal{L}_{\text{TAS}}(K_a, K_s)$, which is the same as the traditional T.-S. scheme. Due to the gaps between heterogeneous students and teachers are not mitigated by our assistant model, the final performance of students is poor too. (3) We remove the supervision from the teacher model to students in Tab. 4 (E), *i.e.*, $\mathcal{L}_{\text{TAS}}(K_t, K_s)$. In this case, although the performance of distilled students is good in some situations, it is not optimal compared to our TAS. This is because the assistant model inevitably damages some knowledge from the teachers, and some easy knowledge is more suitable to transfer by the T.-S. scheme without a middle bridge. Totally, the novel paradigm is powerful for heterogeneous knowledge transfer in cross-architecture distillation.

**Feature and Logit Loss.** We remove the feature loss $\mathcal{L}_{\text{InfoNCE}}$ in Tab. 4 (F) and the logit loss $\mathcal{L}_{\text{OFA}}$ in Tab. 4 (G), and different teacher-student pairs have different performance drops on CIFAR100 and ImageNet-1K. For instance, in CIFAR100, $\mathcal{L}_{\text{OFA}}$ is more important when the teacher is Swin-T and the student is ResNet18, while $\mathcal{L}_{\text{InfoNCE}}$ is more important when the teacher is ConvNeXt-T and the student is Swin-P. In other words, our $\mathcal{L}_{\text{InfoNCE}}$ and $\mathcal{L}_{\text{OFA}}$ are complementary for different teacher-student pairs, so utilizing them jointly will enhance the effectiveness of various distillations. Besides, we demonstrate that MSE loss is not suitable for any T.-S. pairs compared to our INFONCE loss (Tian et al., 2020) in Sec. 4.2 and Appendix E.

Table 5: **Different assistants.** The student is ResNet18 in CIFAR100. $S_c^{1\to2} \to S_m^{3\to fc}$ denotes the assistant is composed of the first two stages of CNN models and the remain parts start from the third stage of MSA models. The others are similar to this definition.

| Teacher | Assistant models with the same length | | | The same CNN modules | |
|---|---|---|---|---|---|
| | $S_c^1 \to S_m^{2\to fc}$ | $S_c^{1\to2} \to S_m^{3\to fc}$ | $S_c^{1\to4} \to S_m^{fc}$ | $S_c^{1\to3} \to S_m^{2\to fc}$ | $S_c^{1\to3} \to S_m^{3\to fc}$ |
| (A): ViT-S | 81.5 / 80.56 | **82.3** / 79.28 | 81.15 / 79.27 | 81.07 / 80.18 | 82.14 / 78.56 |
| | The same MSA modules | | | Ours | |
| (B): Swin-T | $S_c^1 \to S_m^{4\to fc}$ | $S_c^{1\to2} \to S_m^{4\to fc}$ | $S_c^{1\to4} \to S_m^{4\to fc}$ | $S_c^{1\to3} \to S_m^{4\to fc}$ | |
| | 80.84 / 80.23 | 81.7 / 80.93 | 80.11 / 79.98 | 81.93 / **81.61** | |

Table 6: The performance of our assistant (A.) is between that of the teacher (T.) and student (S.).

| Teacher | Student | From Scratch | | TAS | | |
|---|---|---|---|---|---|---|
| | | T. | S. | T. | A. | S. |
| Swin-T | ResNet18 | 81.38 | 69.75 | 81.38 | **76.91** | 72.21 |
| Mixer-B/16 | MobileNetV2 | 76.62 | 68.87 | 76.62 | **73.85** | 72.31 |
| Mixer-B/16 | DeiT-T | 76.62 | 72.17 | 76.62 | **76.30** | 75.00 |
| ConvNeXt-T | ResMLP-S12 | 82.05 | 76.65 | 82.05 | **81.20** | 77.33 |

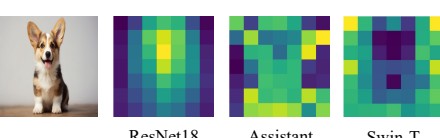

ResNet18     Assistant     Swin-T

Figure 5: The final spatial distribution of the T., A., and S. is different. So we smooth them for feature alignments in $\mathcal{L}_{\text{InfoNCE}}$.

## 4.4 DISCUSSION

**Different components of the assistant model.** We compare the performance when we use different modules of students and teachers to compose our assistant model in Tab. 5. Specifically, we conduct nine different connections between the student ResNet18 and the teacher (A) ViT-S / (B) Swin-T on CIFAR100. As shown in Tab. 5, the best result is 82.3% when the teacher is ViT-S and the assistant model is $S_c^{1\to2} \to S_m^{3\to fc}$ and is 81.61% when the teacher is Swin-T and the assistant model is $S_c^{1\to3} \to S_m^{4\to fc}$. For the hybrid assistant model, CNN modules at the beginning are feature extractors, and MSA modules at the end are feature aggregators, which are complementary and both play important roles (Park & Kim, 2021; Dai et al., 2021). Therefore, although different T.-S pairs have different optimal assistants with different connections, we add an MSA/MLP stage following three CNN stages for the assistant model for simplicity, *i.e.*, $S_c^{1\to3} \to S_m^{4\to fc}$, in most situations.

**The performance and features of the assistant model.** Firstly, as shown in Tab. 6, our assistant model delivers superior performance compared to student models while falling short of the teacher's performance, thereby demonstrating its role as a bridge. Secondly, as shown in Fig. 5, the final features of our assistant model are derived from local CNN models and converted into global receptive fields, *i.e.*, our assistant model combines the knowledge from different inductive biases and module functions in final feature spaces. Lastly, the features of the student, assistant, and teacher models are spatially different in Fig. 5, so it is reasonable to smooth them before transfering in Eq. (3).

## 5 CONCLUSION

**Limitations and Future Works.** (1) It is noteworthy that for certain specific models, such as the extensively studied ResNet18, the performance resulting from distillation by a heterogeneous teacher is inferior to that achieved by a homogeneous teacher. Although our current focus is on the generalizability of various heterogeneous models, yielding significant performance improvements in CAKD, a promising avenue for future research could involve additional prior when specific teacher-student pairs are predefined. (2) Our TAS may disrupt heterogeneous features' spatial alignments. This limitation could be mitigated by aligning extra spatial-level distributions (rather than pixel-level). (3) We have validated our methodology on widely used CV baselines and datasets. While our approach naturally extends to other domains, such as NLP, validating our method across a broader range of domains remains a subject for future investigation.

**Conclusion.** This paper introduces a novel Teacher-Assistant-Student (TAS) scheme designed to enhance the efficacy of heterogeneous distillation. TAS integrates diverse inductive biases and module functions by introducing a hybrid assistant model comprising CNN/MSA/MLP modules derived from students and teachers, thereby improving the feature transfer among heterogeneous models. Extensive experiments demonstrate our TAS is more powerful than other homogeneous and heterogeneous methods on CIFAR100 and ImageNet-1K.

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

## A    IMPLEMENTATION DETAILS

For training models of various architectures on the ImageNet-1K and CIFAR100 datasets, we use different optimization settings and hyperparameters for CNN and MSA/MLP students following the code or paper of OFA (Hao et al., 2023). The detailed settings can be found in Tab. 7. Our code and models are from Timm library (Wightman, 2019). Given that the training pipeline for VisionMamba (Zhu et al., 2024) is currently not integrated into the timm library (Wightman, 2019), we don't consider it following OFA (Hao et al., 2023).

Besides, we set $\gamma$ in $\mathcal{L}_{\text{OFA}}$ as 1.0 on CIFAR100 (Krizhevsky et al., 2009) and 1.5 in ImageNet-1K (Deng et al., 2009) like OFA (Hao et al., 2023). The $\tau_2$ is learnable in $\mathcal{L}_{\text{InfoNCE}}$. The weight of $\mathcal{L}_{\text{OFA}}$ and $\mathcal{L}_{\text{InfoNCE}}$ are equal. We averagely divide the model into 4 stages if the model is not 4-stage in this paper.

If the student and teacher are heterogeneous models, the assistant is:

$$p_{\text{a}}(x) = fc_{\text{m}} \circ \text{S}_{\text{m}}^4 \circ \overbrace{(\text{MSA} \circ \text{PE})}^{\text{L2G}} \circ \text{S}_{\text{c}}^3 \circ \text{S}_{\text{c}}^2 \circ \text{S}_{\text{c}}^1(x), \tag{4}$$

Table 7: **Details of optimization settings.** The settings are following OFA (Hao et al., 2023).

| | CIFAR100 | | ImageNet-1K | |
|---|---|---|---|---|
| | CNN | MSA/MLP | CNN | MSA/MLP |
| Epochs | 100 | 300 | 300 | 300 |
| Image resolution | $224^2$ | $224^2$ | $224^2$ | $224^2$ |
| Batch size | 512 | 1024 | 1024 | 512 |
| Initial LR | 0.1 | 5e-4 | 0.1 | 5e-4 |
| Minimum LR | 1e-6 | 1e-6 | 1e-3 | 1e-5 |
| Optimizer | SGD | AdamW | SGD | AdamW |
| Weight decay | 1e-4 | 5e-2 | 2e-3 | 5e-2 |
| LR schedule | $\times 0.1$ at [30,60,90] | Cosine | Cosine | Cosine |
| Warmup | 3 | 20 | 3 | 20 |
| EMA | - | 0.99996 | - | - |
| RandAugment | - | 9/0.5 | - | 9/0.5 |
| Mixup | - | 0.8 | - | 0.8 |
| Cutmix | - | 1.0 | - | 1.0 |
| RE prob | - | 0.25 | - | 0.25 |

where $x$ is the input image, $S_c^i$ denotes CNN models, $S_m^i$ denotes MSA/MLP models, and $fc_m$ denotes the fully-connected layers of MSA/MLP models.

If the student and teacher are both CNN/MSA/MLP models, the assistant is:

$$p_a(x) = fc_t \circ S_t^4 \circ \overbrace{(\text{MSA} \circ \text{PE})}^{\text{L2G}} \circ S_s^3 \circ S_s^2 \circ S_s^1(x), \tag{5}$$

where $x$ is the input image, $S_s^i$ denotes student models, $S_t^i$ denotes teacher models, and $fc_t$ denotes the fully-connected layers of teacher models.

## B  COMPARISONS WITH OTHER METHODS

We compare the differences between some similar methods and our TAS in Fig. 2. Firstly, to the best of our knowledge, our TAS is the first to apply three-level teacher-assistant-student scheme, which provides more flexible designs for knowledge transfer than existing two-level scheme. Secondly, our assistant model bridges the representation gaps between cross-architecture students and teachers by combining different inductive biases and module functions, making our TAS more suitable for cross-architecture distillation. Thirdly, as demonstrated in (Hao et al., 2023), the $\mathcal{L}_{\text{OFA}}$ enhances the target information and hinders the transfer of incorrect information from the teacher by a modulating parameter $\gamma$ (Hao et al., 2023), which is more suitable than $\mathcal{L}_{\text{KL}}$ in cross-architecture distillation. Lastly, the $\mathcal{L}_{\text{MSE}}$ aligns the features in a pixel-by-pixel manner, which is not reasonable for spatially different heterogeneous features, *e.g.*(A) and (E) in Fig. 6. Thus, as demonstrated in Tab. 10, we get the better performance by smoothing the features in spatial and apply contrastive learning by $\mathcal{L}_{\text{InfoNCE}}$ to align the feature embeddings of cross-architecture models.

There are some works to input the student features to teachers in similar-architecture distillations, *e.g.*, ReviewKD (Chen et al., 2021), FCFD (Liu et al., 2023), and so on (Chen et al., 2022b; Li et al., 2020). However, they are all designed for teacher-student pairs with similar architectures, suggesting different motivations and designs compared to our TAS for cross-architecture distillation. For example, FCFD (Liu et al., 2023) has the best performance and is the most similar method to our TAS, but some important designs of our TAS are very different from FCFD.

Firstly, FCFD (Liu et al., 2023) is designed for CNN students and teachers, which needs to be modified seriously if we apply it to heterogeneous distillations. Besides, as demonstrated in Tab. 8, FCFD is not suitable for any cross-architecture teacher-student models. But our TAS is generic for any teacher-student pair.

Secondly, although FCFD (Liu et al., 2023) also combines different module functions, the connections between students and teachers are **random and mutual**, which makes it hard to converge to the optimal spaces and brings huge training costs, especially for cross-architecture distillations.

Table 8: **Results of FCFD in cross-architecture distillations on CIFAR100 dataset.** As shown, FCFD is not suitable for cross-architecture distillations compared to our TAS.

| Methods | T. | S. | T. | S. | T. | S. |
|---------|----|----|----|----|----|----|
|         | Swin-T | ResNet18 | ViT-S | ResNet18 | ConvNeXt-T | Swin-P |
| FCFD    | 78.34 | | 53.58 | | 77.29 | |
| Our TAS | **81.61** | | **81.93** | | **80.34** | |

Conversely, our TAS considers that the CNN models are feature extractors and the MSA/MLP models are feature aggregators (Park & Kim, 2021), so the assistant is the CNN-MSA/MLP model and obeys the rule of "alternately replacing Conv blocks with MSA blocks from the end of a baseline CNN model" in (Park & Kim, 2021). For example, FCFD (Liu et al., 2023) includes the multiply random connections of MSA/MLP-CNN models (the first parts are MSA/MLP modules, and the latter parts are CNN modules) when we modify it to cross-architecture distillation. However, MSA/MLP-CNN models are unreasonable for the hybrid models (Park & Kim, 2021), leading to bad distillation performance.

Thirdly, FCFD (Liu et al., 2023) is a two-level paradigm that only considers the knowledge transfer between the teacher and student, not introducing the knowledge transfer between the assistant and student. However, the supervision of the assistant is very important as demonstrated in our ablation study of the main paper.

Fourthly, FCFD (Liu et al., 2023) is designed for CNN models and does not consider the representation gaps between different inductive biases between cross-architecture models. Thus, the feature projectors of FCFD are CNN modules, which perform worse than our L2G modules because L2G includes the MSA modules to convert the local features to global receptive fields.

Lastly, FCFD (Liu et al., 2023) does not consider the gaps between different representation spaces of cross-architecture models. Thus, the loss functions of FCFD (Liu et al., 2023), *i.e.*, $\mathcal{L}_{\mathrm{KL}}$ and $\mathcal{L}_{\mathrm{MSE}}$ are not appropriate for cross-architecture teacher-student pairs. For example, as demonstrated in Tab. 10, $\mathcal{L}_{\mathrm{MSE}}$ is not suitable for some cross-architecture teacher-student pairs.

Experimentally, as shown in Tab. 3, our TAS has a competitive performance compared with FCFD (Liu et al., 2023) in similar-architecture distillations. More importantly, our TAS is generic for cross-architecture distillations, but FCFD is hard to achieve it in Tab. 8.

## C  HETEROGENEOUS FEATURES

Fig. 6 shows heterogeneous features, demonstrating some important observations in our main paper.

Firstly, heterogeneous models have different inductive biases. For example, CNN models (He et al., 2016) have the inductive bias of "locality", thereby making the features local like (A-B) in Fig. 6. Differently, the features of MSA and MLP models (Liu et al., 2021b; Dosovitskiy et al., 2021; Tolstikhin et al., 2021) are global because of their global inductive bias, *e.g.*, (C-F) in Fig. 6. Therefore, combining different inductive biases helps mitigate the gaps between heterogeneous models like our assistant.

Secondly, heterogeneous models have different module functions. For example, the architectures/functions of ResNet (He et al., 2016) and Swin models (Liu et al., 2021b) are hierarchical. They gradually expand the receptive fields and upsample the features, *e.g.*, (A-D) in Fig. 6. Differently, MLP models (Tolstikhin et al., 2021) and most MSA models (Dosovitskiy et al., 2021) are uniform. The features of shadow and deep layers have higher similarity than the hierarchical CNN, *e.g.*, (E-F) in Fig. 6. Therefore, combining different module functions helps mitigate the gaps between heterogeneous models like our assistant.

Thirdly, features of heterogeneous models have different spatial distributions in different channels. For example, the different channels of CNN models have similar spatial localizations (such as the right figures of Fig. 6 (A-B)). Conversely, the features of MSA and MLP models in different channels are more diverse, *e.g.*, the right figures of Fig. 6(C-F). Besides, as demonstrated in (Park & Kim, 2021; He et al., 2016; Tolstikhin et al., 2021), spatial smoothing is useful for the predictions of CNN/MSA/MLP models (*e.g.*, average pooling). Therefore, we smooth the features and replace

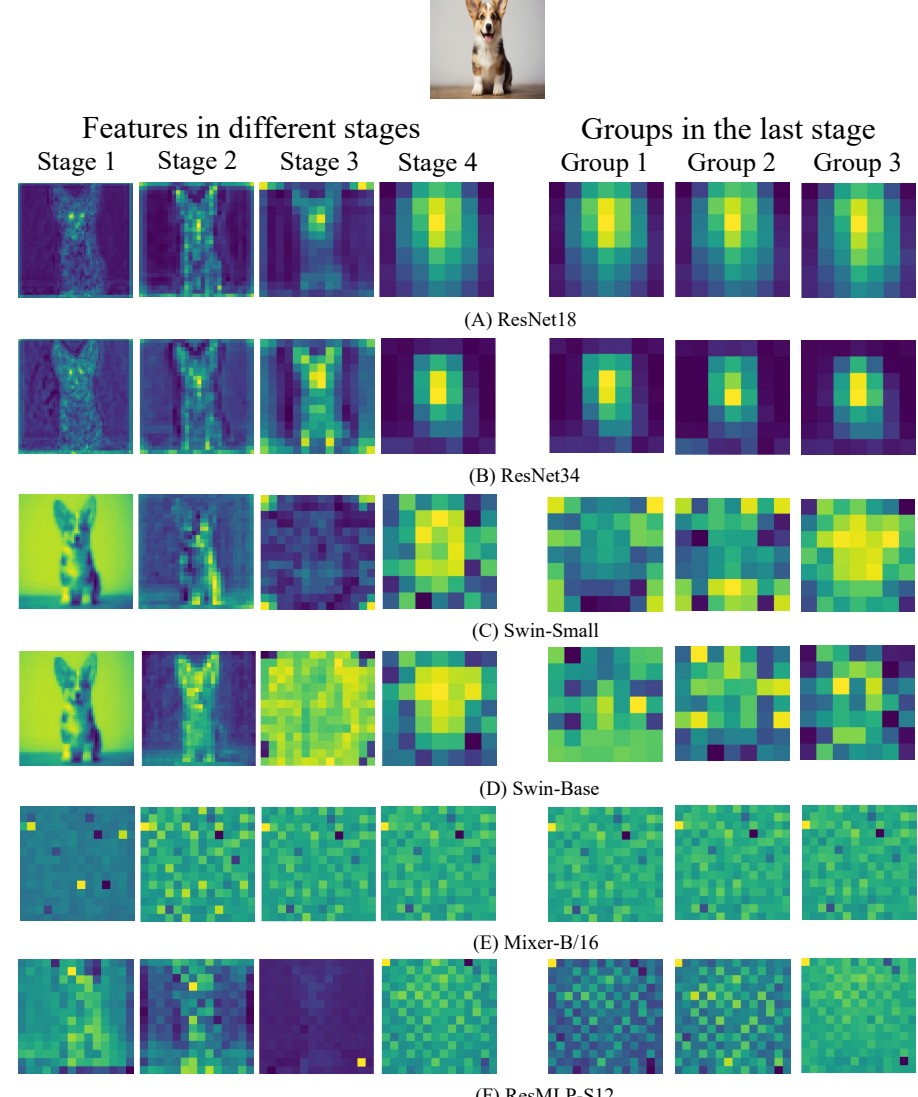

Figure 6: **Diverse features in different models.** The left figures are features in different stages (all models are divided into 4 stages). The right figures are the final features in different groups (we divide the channels of final features into 3 groups). The spatial distribution of features is diverse according to the channels, stages, and model architectures/functions.

pixel-by-pixel $\mathcal{L}_{\text{MSE}}$ with $\mathcal{L}_{\text{InfoNCE}}$ in our main paper. Tab. 10 also demonstrates the strength of applying $\mathcal{L}_{\text{InfoNCE}}$ to smoothing features.

## D    DIFFERENT DISTILLING STRATEGY

Multi-teacher progressive distillation is a training strategy and our TAS is a training algorithm. They are **orthogonal** and can be used together. As shown in (D-E) in Tab. 9, we can replace the T.-S. with our T.-A.-S. paradigm to improve the results in each stage of progressive distillation. Besides, using multi-teacher distillation, we can improve the performance of a given student by applying our TAS to both SAKD and CAKD.

Table 9: **Different distillation paradigm.** Swin denotes the Swin-Tiny model. Our method is the one-stage joint-optimization teacher-assistant-student paradigm, which is orthogonal with progressive distillation like (Cao et al., 2023; Mirzadeh et al., 2020).

| Methods (all methods only use the $\mathcal{L}_{\mathrm{KD}}$) | CIFAR100 |
|---|---|
| (A) Swin → ResNet34 + Resnet34 → ResNet18 | 78.53 |
| (B) Swin-Assistant-ResNet18 (ours w/o $\mathcal{L}_{\mathrm{TAS}}(\mathrm{K_t}, \mathrm{K_s})$) | 79.26 |
| (C) Swin-Assistant-ResNet18(ours w/ $\mathcal{L}_{\mathrm{TAS}}(\mathrm{K_t}, \mathrm{K_s})$) | 79.28 |
| (D) Swin → ResNet18 + ResNet34 → ResNet18 | 80.07 |
| (E) Swin-Assistant-ResNet18 + ResNet34-Assistant-ResNet18 | 81.24 |

## E  LOSS FUNCTIONS FOR CROSS-ARCHITECTURE DISTILLATIONS.

In Tab. 10, we compare the results of FitNet with different settings on the CIFAR100 dataset. Firstly, the accuracy improves from 24.06 to 65.17 when we apply $\mathcal{L}_{\mathrm{MSE}}$ only to features of the final stage, rather than intermediate stages. This demonstrates the intermediate features are not suitable for feature alignment in some cross-architecture teacher-student pairs. Secondly, the accuracy improves from 65.17 to 76.79 when we apply average pooling to features of the final stage. This demonstrates the diverse spatial distributions of features are not suitable for feature alignment in some cross-architecture teacher-student pairs. Thirdly, the accuracy improves from 76.79 to 78.01 when we replace $\mathcal{L}_{\mathrm{MSE}}$ with $\mathcal{L}_{\mathrm{InfoNCE}}$. This is because $\mathcal{L}_{\mathrm{InfoNCE}}$ considers the releationships between different channels, but $\mathcal{L}_{\mathrm{MSE}}$ is pixel-by-pixel.

Table 10: **FitNet (Romero et al., 2015) with $\mathcal{L}_{\mathrm{MSE}}$ *vs.* $\mathcal{L}_{\mathrm{InfoNCE}}$ loss on CIFAR100.** The teacher is ConvNeXt-T (88.41% Top-1 accuracy) and the student is Swin-P (72.63% Top-1 accuracy) on CIFAR100. As shown, the smoothing features and $\mathcal{L}_{\mathrm{InfoNCE}}$ are more suitable for cross-architecture distillations than the original features and $\mathcal{L}_{\mathrm{MSE}}$

| Loss | Top-1 accuracy |
|---|---|
| $\mathcal{L}_{\mathrm{MSE}}$ in all intermediate features | 24.06 |
| $\mathcal{L}_{\mathrm{MSE}}$ in the final features | 65.17 |
| $\mathcal{L}_{\mathrm{MSE}}$ in the final features after average pooling | 76.79 |
| $\mathcal{L}_{\mathrm{InfoNCE}}$ in the final features after average pooling | **78.01** |

## F  L2G IN OUR ASSISTANT MODEL.

As shown in Tab. 11, when we replace the Swin block (Liu et al., 2021b) with ViT block (Dosovitskiy et al., 2021) in our L2G, the performance on different teacher-student pairs has different rises and falls. Thus, the MSA block in L2G is also important and is worth exploring in future works.

Our L2G includes a patch embedding for dimension alignments of features and an MSA block for global information exchange.

*Why do we use a patch embedding?* The feature shape of a CNN model with the size (N, C, H, W), while that of an MSA/MLP model is denoted as (N, L, D). N indicates the batch size, and C, H, and W refer to the channel, height, and width of the CNN model's feature map respectively. L and D denote the patch number and embedding dimension of the ViT/MLP model's feature map. In our assistant model, to connect the features of CNN and MSA/MLP models, we need to transform the feature map of the CNN model into the MSA/MLP-style (shape) feature through a "patchify" operation. Besides, the effectiveness of "divide image to patches" has been demonstrated in CNN models (Trockman & Kolter, 2022), MSA models (Dosovitskiy et al., 2021; Liu et al., 2021b), and MLP models (Li et al., 2023a; Touvron et al., 2022; Tolstikhin et al., 2021). Therefore, we use a patch embedding to process the CNN features. More importantly, patch embedding is a kind of spatial smoothing that is beneficial to align heterogeneous features.

Table 11: **Our results with different MSA blocks.** T. and S. denote the teacher and the student. ViT block is from ViT (Dosovitskiy et al., 2021) and Swin block is from Swin (Liu et al., 2021b). As shown, different blocks have different functions in different teacher-student pairs.

| MSA Block | T. | S. | T. | S. | T. | S. |
|---|---|---|---|---|---|---|
| | Swin-T | ResNet18 | ViT-S | ResNet18 | ConvNeXt-T | Swin-P |
| ViT Block | 81.05 | | 80.74 | | **80.72** | |
| Swin Block | **81.61** | | **81.93** | | 80.34 | |

*Why do we use an MSA block?* The extra MSA block converts the local CNN features to a global receptive field, which is more suitable to input the later MSA/MLP models. Besides, the later MSA/MLP models are frozen when the teacher is the MSA/MLP model and the student is the CNN model. In this case, a learnable MSA block plays an important role in aligning heterogeneous features.

Table 12: **Training cost.** The extra parameters of our TAS are about one-tenth of OFA (Hao et al., 2023). The best results are **bold**.

| Teacher | Student | Student Params | Student FLOPs | OFA Branch | | Our Branch | |
|---|---|---|---|---|---|---|---|
| | | | | Params | FLOPs | Params | FLOPs |
| DeiT-T | ResNet18 | 11.69 M | 1.82 G | 4.92 M | 0.1 G | **0.39 M** | **0.07 G** |
| ResNet50 | DeiT-T | 5.68 M | 1.08 G | 5.81 M | 0.25 G | **0.54 M** | **0.10 G** |
| ConvNeXt-T | ResMLP-S12 | 15.32 M | 3.01 G | 21.64 M | 0.99 G | **1.48 M** | **0.29 G** |

# G    TRAINING COST.

Beyond performance considerations, training cost is critical for the distillation. We compare the training cost of the recent OFA (Hao et al., 2023) and our TAS framework in Tab. 12. In Fig. 4(f), OFA uses four extra feature projectors, but our TAS only uses one L2G to project the features at the third stage of CNN models. Therefore, as shown in Tab. 12, we introduce much fewer additional parameters and FLOPs on par with OFA (Hao et al., 2023) under different combinations of teacher and student models. Specifically, the number of extra parameters is about one-tenth that of OFA when the student and teacher are different architectures. As a result, TAS is more efficient than OFA (Hao et al., 2023).

