# OpenReview forum: "TAS: Distilling Arbitrary Teacher and Student via a Hybrid Assistant"
_ICLR.cc/2025/Conference — ICLR 2025 Conference Withdrawn Submission_

### Official Review · Reviewer_kjk5 · 2024-10-17

**Soundness:** 2
**Presentation:** 2
**Contribution:** 2
**Rating:** 3
**Confidence:** 4

**Summary:**

This paper presents an approach to facilitating cross-architecture knowledge distillation by employing a carefully designed teaching assistant structure. The assistant model combines different inductive biases and utilizes appropriate translation modules to ensure knowledge is transferred in a way that maximizes distillation efficiency. The authors provide experimental results using various combinations of teachers, students, and datasets, demonstrating that the proposed method can outperform the OFA approach.

**Strengths:**

+ Extensive experimental validation, covering a wide range of different architectures and models

+ Conducted experiments lead to consistent improvements

+ Ablation studies provided

**Weaknesses:**

- The authors give the impression that Teacher Assistants (TA) are being introduced for the first time in this paper. This is suggested early in the manuscript, where they state: "To alleviate heterogeneous feature gaps in CAKD, we introduce a Teacher-Assistant-Student distillation paradigm (T.-A.-S.1, called TAS) by incorporating a hybrid assistant model as a bridge to facilitate smoother knowledge transfer"
I am unsure whether this is an oversight or if the authors are trying to emphasize a different aspect of their approach. However, presenting this as a main claim is problematic, as Teacher Assistants were already introduced in 2020 with the specific aim of mitigating the distillation gap.

I am even more surprised that the seminal work of Mirzadeh et al. (Mirzadeh, Seyed Iman, et al. "Improved knowledge distillation via teacher assistant." Proceedings of the AAAI Conference on Artificial Intelligence, 2020) is only referenced in the related work section. This work is conspicuously absent from the provided taxonomy, despite being the most relevant in my opinion.

Other important studies that have addressed distillation between heterogeneous architectures are not discussed either, e.g., Liu, Yufan, et al. "Cross-architecture knowledge distillation." Proceedings of the Asian conference on computer vision. 2022.,


- The lack of appropriate comparisons with recent literature on teacher assistants makes it difficult to evaluate the true novelty of the paper. This gap is also apparent in the experimental comparisons, where the authors only compare their method with one approach (OFA) that addresses the issue of heterogeneous distillation, without including any other techniques that employ teacher assistants. Although the authors demonstrate significant performance improvements, I cannot adequately assess the importance of their results due to the lack of relevant comparisons. This is another critical weakness.

- While the authors attempt to motivate their approach by examining the potential impact of inductive biases on distillation, I am uncertain that the paper offers truly valuable insights in this regard. The proposed architectural technique may be effective, but in my view, it does not represent an ICLR-level novelty.

- Additionally, the paper suffers from some repetition, which makes it somewhat difficult to follow.

For example, in page 2 module functions are defined "from module functions, i.e., how the models will read, decode, and process the input features. "
and then in p.4 module functions are re-introduce "Module functions describe how a model reads, encodes, decodes, and processes the data (Liu et al., 2023)"
This is just one instance of several repetitive statements throughout the manuscript. While not a major factor in my decision, these repetitions weaken the overall quality of the paper.

Overall, I cannot recommend this paper for publication in ICLR due to potentially misleading presentation, lack of appropriate comparisons and incremental contribution.

**Questions:**

- Please clarify whether the main contribution of the paper is the introduction of "a Teacher-Assistant-Student distillation paradigm" and how this paradigm relates to existing Teacher Assistant distillation frameworks. If there is any additional novelty beyond this, it should be clearly explained, as the current presentation leaves the impression of reintroducing an existing approach without proper acknowledgment.

- Additionally, please clarify why no comparisons with other teacher assistant approaches have been included. The absence of such comparisons makes it difficult to evaluate the significance and novelty of the proposed method.

---

### Official Review · Reviewer_mAzj · 2024-10-29

**Soundness:** 2
**Presentation:** 2
**Contribution:** 2
**Rating:** 5
**Confidence:** 3

**Summary:**

The TAS method introduces a hybrid assistant model to bridge the feature gap between heterogeneous teacher and student models in Cross-Architecture Knowledge Distillation (CAKD). The assistant model merges convolutional and attention-based modules, creating a versatile intermediary that accommodates both local and global feature representations. This approach improves knowledge transfer by aligning heterogeneous inductive biases and module functions, aided by a spatial-agnostic InfoNCE loss to address spatial feature distribution disparities. TAS enhances the flexibility and performance of knowledge distillation across diverse architectures, achieving notable accuracy improvements on datasets like CIFAR-100 and ImageNet-1K.

**Strengths:**

I can consider the strengths of these paper as following:

1) Versatile Distillation for Heterogeneous Models: TAS enables effective knowledge transfer across different architectures, including CNNs, ViTs, and MLPs, by bridging feature gaps, a common challenge in heterogeneous distillation.

2) Hybrid Assistant Model: Incorporating both teacher and student module functions allows the assistant to effectively handle feature discrepancies due to varying inductive biases, enhancing distillation performance.

3) Spatial-Agnostic Feature Alignment: The use of InfoNCE loss for feature alignment accommodates spatial diversity in features, unlike traditional MSE loss, which struggles with spatially diverse architectures.
4) Superior Performance Gains: TAS consistently improves student model performance, with reported gains of up to 11.47% on CIFAR-100 and 3.67% on ImageNet-1K, demonstrating its effectiveness across benchmarks.
Lower Training Cost: The framework introduces minimal additional learnable parameters, reducing the overhead while achieving competitive results.

**Weaknesses:**

Overall, I found the paper somewhat difficult to follow. For example, the text references images located 2-3 pages later (e.g., line 71), which disrupts the flow.

Regarding technical aspects, I noted the following weaknesses:

1) Complexity and Resource Demands: While TAS reduces training costs, it still requires substantial resources. This is due to the three-level knowledge transfer paradigm and the hybrid assistant model, which increase complexity.

2) Dependence on Careful Model Design: The success of the assistant model relies heavily on how effectively it integrates the functions of the teacher and student modules. This dependency could limit its adaptability to specific model architectures.

3) Risk of Overfitting with Small Datasets: TAS demonstrates strong performance with large datasets; however, its complexity could lead to overfitting or reduced benefits on smaller or less diverse datasets.

4) Increased Complexity in Loss Function: The use of InfoNCE and OFA losses improves feature alignment but adds tuning complexity. This may affect ease of implementation compared to simpler knowledge distillation methods.

**Questions:**

1) Similarity to Mixture of Experts: The TAS framework bears resemblance to the Mixture of Experts (MoE) method, particularly in its use of a multi-level knowledge transfer paradigm and the integration of teacher and student functions. Could you elaborate on the distinctions between TAS and MoE, especially regarding model architecture and training dynamics? Additionally, what specific advantages does TAS offer over MoE, and under what conditions might one method be preferable to the other?

2) Resource Efficiency: The TAS framework claims reduced training costs yet introduces a multi-level knowledge transfer and a hybrid assistant model, which appear resource-intensive. Could you provide more details on the actual resource savings compared to traditional knowledge distillation (KD) methods? Additionally, how scalable is TAS when applied to resource-constrained environments?

3) Generalization on Smaller Datasets: While TAS shows robust improvements on large datasets, there is potential for overfitting on smaller datasets. Have you evaluated TAS's performance across different dataset sizes and diversities? What mitigation strategies would you suggest for reducing overfitting risks in such cases?

4) Model Design Dependency: The performance of the assistant model depends significantly on effective integration of teacher and student module functions. Can you elaborate on how TAS might be adapted or optimized for models with varying architectures? Would it be feasible to use TAS in scenarios where teacher-student model compatibility is less certain?

5) Complexity of Loss Functions: While InfoNCE and OFA losses enhance feature alignment, they add tuning complexity. Could you discuss the tuning process and the sensitivity of TAS’s performance to different hyperparameter settings? How would you assess the trade-off between the additional alignment benefits and the added implementation complexity compared to simpler KD approaches?

---

### Official Review · Reviewer_zvvo · 2024-11-02

**Soundness:** 3
**Presentation:** 3
**Contribution:** 2
**Rating:** 3
**Confidence:** 5

**Summary:**

This paper proposes an assistant model as a bridge to facilitate smooth feature knowledge transfer between heterogeneous teachers and students. It combines the advantages of cross-architecture inductive biases and module functions by merging convolution and attention modules derived from both student and teacher module functions. Experimental results on image classification tasks across many network architectures show the effectiveness of the proposed method.

**Strengths:**

1.	This paper is well-written and easy to understand.
2.	The motivation of this paper is clear. The homogeneous distillation often suffers from limited potential and flexibility.
3.	This paper conducts many analyses among CNN/MSA/MLP models. Therefore, merging convolution and attention modules derived from both student and teacher module functions for CAKD is reasonable.
4.	Experimental results on image classification tasks across many network architectures show the effectiveness of the proposed method.

**Weaknesses:**

1.	The proposed framework is incremental, since the assistant-based KD framework has been adopted in the previous work [1].
2.	The proposed method seems working for CAKD. This paper does not explain how the proposed TAS works for SAKD.
3.	The paper posts the performance of teachers, assistants and students in Table 6, but does not give a comparison of model sizes among the teacher model, the assistant model and the student model, which is important in judging knowledge distillation tasks.
4.	For the Res34-Res18 pair on imagenet, some previous distillation works achieve stronger results than this paper ‘s 72.29. For example, channel distillation [2] reports 72.39, and DiffKD [3] reports 72.49. Therefore, it is may be overclaimed that the proposed method “achieving state-of-the-art performance”.
5.	Experimental results are only conducted on image classification tasks. More downstream experiments such as object detection and semantic segmentation should be conducted to demonstrate the effectiveness of the proposed method.

Writing suggestion: In the explanation of the assistant model, the use of the operation symbol ◦ in equation (2) is not explained. This problem also appears in the formulas in the appendices.

Reference:

[1] Mirzadeh S I, Farajtabar M, Li A, et al. Improved knowledge distillation via teacher assistant[C]//Proceedings of the AAAI conference on artificial intelligence. 2020, 34(04): 5191-5198.

[2] Zhou Z, Zhuge C, Guan X, et al. Channel distillation: Channel-wise attention for knowledge distillation[J]. arXiv preprint arXiv:2006.01683, 2020.

[3] Huang T, Zhang Y, Zheng M, et al. Knowledge diffusion for distillation[J]. Advances in Neural Information Processing Systems, 2024, 36.

**Questions:**

Please see the Weakness section.

---

### Official Review · Reviewer_AQhS · 2024-11-03

**Soundness:** 2
**Presentation:** 2
**Contribution:** 2
**Rating:** 5
**Confidence:** 3

**Summary:**

The authors propose a new cross-architecture knowledge distillation method named TAS. They find that different models have much different features in heterogeneous distillation because of different inductive biases and module functions. Therefore, TAS uses an assistant model to help the teacher models to transfer the knowledge to student models smoothly, which connects the modules of CNN and MSA/MLP with a MSA projector. In addition, TAS adopts InfoNCE loss and OFA loss to supervise the transfer of features and logits respectively. Compared with other distillation methods, TAS achieves better performance in most situations on the CIFAR-100 dataset and ImageNet-1K dataset.

**Strengths:**

(1) The motivation is well demonstrated.

(2) TAS demonstrates better performance compared with other distillation methods in most situations.

(3) The authors provide sufficient ablative results to study the components of TAS.

**Weaknesses:**

(1) Need to validate with larger student models. Following the experimental settings of OFA [1], TAS selects the ResNet18, MobileNetV2, DeiT-T, Swin-N and ResMLP-S12 as the student models for main experiments, as shown in Table 1 and Table 2. All of the above models are relatively light-weight. In the table 6 of OFA [1], it provides the results with ResNet152|ViT-B as the teacher and ResNet50 as the student. Besides OFA, there are also some works which provide the distillation results between heterogeneous teachers and students. For example, both [2] and [3] provide experiment results with Swin-L as the teacher and ResNet-50|Swin-T as the student. I think the authors should conduct experiments with larger student models to validate the generalization ability of TAS.

[1] Hao, Zhiwei, et al. "One-for-all: Bridge the gap between heterogeneous architectures in knowledge distillation." NeurIPS, 2023.

[2] Huang, Tao, et al. "Knowledge distillation from a stronger teacher." NeurIPS, 2022.

[3] Huang, Tao, et al. "Knowledge diffusion for distillation." NeurIPS, 2023.

(2) While the main motivation of TAS is to introduce an assistant model as a bridge to facilitate smooth feature knowledge transfer between heterogeneous teachers and students, the description for the design of assistant model only focuses on the distillation between CNN students and MSA/MLP teachers, which uses a local-to-global (L2G) feature projector. In Table 1 and Table 2, there are some results for distillation between CNN teachers and MSA/MLP students. I think the authors should explain why the design can still work in the situation. In the story, a global-to-local feature projector sounds more suitable.

(3) Not very important. The painting of the Figure 3 should be improved. For example, in Figure 3 (c), we can observe white border and some very short black line for each feature map. This is probably due to screenshot. Figure 3 (b) doesn't look good compared with other figures because of the while line used to demonstrate pachifying and the colors. While it's not important.

**Questions:**

(1) I find that InfoNCE Loss plays an important role in the design of TAS, as shown in Table 4 (F). When removing the InfoNCE Loss, TAS can not get better performance compared to the OFA in ImageNet. In my opinion, OFA is the major counterpart for TAS. If we add the InfoNCE Loss to the OFA after the global average pooling layer at each branch (or only the global average pooling layer of the student head), will it achieve better performance than TAS?

(2) Compared to OFA, I think the major advantage of TAS is the training cost since OFA introduces multiple heavy branches for distillation. Besides the flops and parameters shown in Table 12, the authors could also evaluate the training time (e.g., A100 hour) and memory cost. While the authors compare the training cost of TAS with other methods in the table of Figure 2, it's not intuitive enough. For example, if OFA performs a better trade-off between accuracy and training time compared to other methods (KD, FitNet, CRD, FCFD, OFA), the authors could use a scatter plot to demonstrate it. If TAS is significantly more efficient than OFA, the authors could emphasize the advantage in the main text with some figures, which will make TAS more attractive.

---

### Note · Authors · 2024-11-13

**Comment:**

Thanks for the efforts of ACs.
Thanks for the valuable reviews of all reviewers.

**Withdrawal Confirmation:**

I have read and agree with the venue's withdrawal policy on behalf of myself and my co-authors.